# A C57BL/6 Mouse Model of SARS-CoV-2 Infection Recapitulates Age- and Sex-Based Differences in Human COVID-19 Disease and Recovery

**DOI:** 10.3390/vaccines11010047

**Published:** 2022-12-25

**Authors:** Michael A. Davis, Kathleen Voss, J. Bryan Turnbull, Andrew T. Gustin, Megan Knoll, Antonio Muruato, Tien-Ying Hsiang, Kenneth H. Dinnon III, Sarah R. Leist, Katie Nickel, Ralph S. Baric, Warren Ladiges, Shreeram Akilesh, Kelly D. Smith, Michael Gale

**Affiliations:** 1Center for Innate Immunity and Immune Disease, Department of Immunology, University of Washington, Seattle, WA 98109, USA; 2Department of Microbiology and Immunology, University of North Carolina at Chapel Hill, Chapel Hill, NC 27514, USA; 3Department of Epidemiology, University of North Carolina at Chapel Hill, Chapel Hill, NC 27514, USA; 4Department of Pathology, University of Washington, Seattle, WA 98195, USA; 5Comparative Medicine, University of Washington, Seattle, WA 98195, USA; 6Rapidly Emerging Antiviral Drug Discovery Initiative, University of North Carolina at Chapel Hill, Chapel Hill, NC 27514, USA; 7Department of Laboratory Medicine and Pathology, University of Washington, Seattle, WA 98195, USA

**Keywords:** SARS-CoV-2, COVID-19, Age, Sex, Inflammation, Innate Immunity, IFN

## Abstract

We present a comprehensive analysis of SARS-CoV-2 infection and recovery using wild type C57BL/6 mice and a mouse-adapted virus, and we demonstrate that this is an ideal model of infection and recovery that phenocopies acute human disease arising from the ancestral SARS-CoV-2. Disease severity and infection kinetics are age- and sex-dependent, as has been reported for humans, with older mice and males in particular exhibiting decreased viral clearance and increased mortality. We identified key parallels with human pathology, including intense virus positivity in bronchial epithelial cells, wide-spread alveolar involvement, recruitment of immune cells to the infected lungs, and acute bronchial epithelial cell death. Moreover, older animals experienced increased virus persistence, delayed dispersal of immune cells into lung parenchyma, and morphologic evidence of tissue damage and inflammation. Parallel analysis of SCID mice revealed that the adaptive immune response was not required for recovery from COVID disease symptoms nor early phase clearance of virus but was required for efficient clearance of virus at later stages of infection. Finally, transcriptional analyses indicated that induction and duration of key innate immune gene programs may explain differences in age-dependent disease severity. Importantly, these data demonstrate that SARS-CoV-2-mediated disease in C57BL/6 mice phenocopies human disease across ages and establishes a platform for future therapeutic and genetic screens for not just SARS-CoV-2 but also novel coronaviruses that have yet to emerge.

## 1. Introduction

With the current SARS-CoV-2 pandemic, humans have now experienced three coronavirus (CoV) outbreaks in the last 20 years (SARS-CoV-1, MERS, SARS-CoV-2), and additional coronavirus outbreaks are expected in the future. Animal model systems are powerful tools to uncover mechanisms of pathology and to identify genetic dependencies for viral disease. Compared to non-human primates, ferrets, and other less widely used rodents, laboratory mice are the gold standard animal models of virus infection and disease course, in part as they permit genetic screens aimed at identifying immune factors and pathways required for disease and recovery. Complicating animal studies though, the ancestral strain SARS-CoV-2 did not naturally infect mice at the time of emergence, which was true for MERS and SARS-CoV-1 as well [1,2]. For SARS-CoV-1 and SARS-CoV-2, this host restriction is due to sequence polymorphisms in murine Ace2 [3], the main receptor that binds viral Spike protein and permits cellular entry. To address this problem, researchers have used transgenic mice expressing human ACE2 under the control of the keratin 18 promoter (K18-Ace2). These mice can be infected with ancestral SARS-CoV-2 and subsequent viral variants wherein they can develop severe pulmonary disease [4,5]. However, due to ectopic expression of ACE2, these mice suffer from lethal neuroinflammation, a feature that is not seen in humans with COVID-19. Moreover, the hAce2 model complicates host genetic screens as knockout strains of interest must first be bred to the K18-Ace2 line. These limitations drove work to develop SARS-CoV-2 infection models in wild type (WT) mice using mouse-adapted (MA) viral variants [3,6,7,8,9,10,11,12,13]. While use of mouse adapted viruses allows infection of wild type mice, it must be noted that these strains harbor mutations that can alter disease pathology such that it no longer resembles the strain of interest. Thus careful studies are needed to verify the effectiveness and suitability of these strains as model organisms. Studies employing mouse-adapted SARS-CoV-2 strains have primarily been performed in BALB/c mice [6,8,9,13,14] and focused on a single sex or single age cohorts [3,10,12,15,16], or have employed highly adapted and virulent variants that limit the ability to comprehensively examine infection and recovery [11].

The SARS-CoV-2 pandemic is moving towards an endemic phase and is likely to remain a persistent threat as the viral genome continues to evolve [17]. Viral evolution of SARS-CoV-2 leads to the continual emergence of new variants of concern replacing previous circulating variants [18], and also led to the ability of SARS-CoV-2 variants to infect mice [19,20,21]. Additionally, current SARS-CoV-2 variants mediate a disease course that is much less pathogenic than the ancestral strain and has transitioned from causing a severe lower respiratory tract infection to a relatively mild upper respiratory tract infection [17]. Despite the lessened pathogenicity of newer SARS-CoV-2 variants, there remains a need to understand the processes underlying infection and recovery. Moreover, it is important to note that despite the potential for deadly outcome of severe COVID-19, ~99% of people infected with SARS-CoV-2 throughout the pandemic recovered from infection with only mild to moderate disease or were asymptomatic (https://covid19.who.int/, accessed on 1 October 2022). Thus, there remains an important need for comprehensive in vivo analyses of SARS-CoV-2 infection models to understand the dynamics of infection and recovery. For example, there is a clear and pronounced increase in disease severity based on age and sex [22,23,24], but it is not fully understood whether these differences are driven by age- and sex-dependent comorbidities (such as obesity) or inherent differences of the host (such as sex-linked expression of immune genes and sex-hormone sensitive immune cells). In vivo models of infection can set the stage for future genetic screens aimed at identifying key immune pathways required for recovery as well as identifying the age- and sex-based factors that affect disease severity. Therefore, the goal of the present study was to develop an immunocompetent mouse model of SARS-CoV-2 infection that displays main features of human infection and recovery, starting with ancestral SARS-CoV-2. Here, we comprehensively describe a C57BL/6 model of SARS-CoV-2 infection and show the utility of this model in examining acute infection, progression to disease, lung disease, and recovery. As a great many transgenic C57BL/6 lines exist that lack specific genes or express modified versions of specific genes, we demonstrate that this model provides a platform for genetic interrogation of the virus/host interactions that control SARS-CoV-2 infection and immunity.

## 2. Materials and Methods

### 2.1. Mice

10 wk-old C57BL/6J mice (Jax #000664) were purchased from Jackson laboratories directly or bred in-house from animals purchased from Jackson laboratories. 20-wk old C57BL/6J mice were purchased from Jackson Laboratories at 10 wks of age and aged to 20 wks old. SCID (B6.Cg-Prkdc^scid^/SzJ, #001913) mice were purchased from Jackson laboratories. 2 yr-old C57BL/6 mice were obtained from the National Institute of Aging subsidized aging rodent colony at Charles River, Inc. Mice were housed with appropriate food, water, and enrichment in individually ventilated cages and were euthanized if they fell below 70% original weight or became otherwise moribund, consistent with previously published studies [3,8]. Weights of all animals studied and numbers of mice per cohort in each study can be found in Appendix A; however, studies of weight loss used 5 mice per cohort and tissue collection studies used 3 mice per cohort unless otherwise noted.

### 2.2. Infections and Monitoring

Mice were anesthetized with Ketamine/Xylazine at roughly 80–100 mg/kg ketamine and 5–10 mg/kg xylazine. Anesthetized mice were inoculated intranasally with indicated doses of virus diluted in saline at a total volume of 50 mL or with 50 mL saline alone as a mock control. Mice were closely monitored following anesthetization until they recovered and then were monitored daily for changes in health and/or weight. Mice were also monitored daily by UW DCM staff to ensure the animals always had sufficient food and water and that cage conditions met approved standards.

### 2.3. Virus

SARS-CoV-2 MA10 was kindly provided by Ralph S. Baric. A P3 stock was amplified to 8 × 10^6^ PFU/mL in VERO (USAMRIID) cells and sequence verified. Sequence analysis indicated an R2G mutation in NSP7.

### 2.4. RNA Prep

Tissue collected at necropsy was originally placed in RNA later (Thermofischer, Waltham, MA, USA, AM7021), stored at 4° for 24 h to one week, and then transferred to −20° for longer storage. To process tissue, samples were transferred to 1 mL Trizol (Thermofischer, 15596018) in 2 mL Percellys hard tissue homogenization tubes (Cayman Chemical Co., Ann Arbor, MI, USA, 10011151). Samples were homogenized in a Percellys 24 Homogenizer using 1–3 cycles of 15” at 6500 rpm and placing samples on ice between cycles. RNA was then purified from the Trizol homogenate using the RNeasy 96 kit (Qiagen, Hilden, Germany), according to manufacturer’s protocols.

### 2.5. Histology & RNA-ISH

For each mouse a sample of lung was incubated in formalin at room temperature for at least seven days to fix tissue and inactivate virus. The fixed tissue was processed and embedded in paraffin. 5 µm sections were cut and stained with hematoxylin and eosin (H&E). Airway pathology was assessed in H&E stained sections to assess bronchial epithelial cell death (score: 0 = no dead cells, 1 = 1–5 dead cells, 2 = 6–10 dead cells, 3 = 11–20 dead cells and 4 = >20 dead cells; scored for ten 400× fields per mouse lung); endarteritis, venulitis, and perivenule, peribronchial and periarterial inflammation (score 0 = none, 1 = 0–25% circumference with >1 leukocyte cell layer, 2 = 26–50% circumference with >1 leukocyte cell layer, and 3 = 50–100% circumference with >1 leukocyte cell layer; scored for 10 400X fields per mouse lung); and interstitial pneumonitis and pulmonary edema (score = percentage of pulmonary alveolar parenchyma with septae expanded by leukocyte; scored for ten 100X fields). Scoring sheets showing scores for individual mice as well as calculations of average scores are found in Appendix A. Importantly, the pathologist was also blinded to samples for analysis.

RNA in situ hybridization (RNA-ISH) was performed on FFPE sections of mouse tissues using the V-nCoV2019-S probe (Catalog# 848561) for sense strand of the Spike gene and the V-nCoV2019-S-sense probe (Catalog# 845701O) for the antisense strand of the Spike gene. RNA-ISH signal was developed using the RNAscope™ 2.5 HD Assay—RED development kit (Catalog # 322350) from ACDBio Inc. (Newark, CA, USA), and slides were counterstained with hematoxylin for tissue visualization. For scoring of infected bronchioles, individual bronchiole profiles were assessed for the percentage of infected cells according to the scheme outlined in Appendix A. Each of these bronchioles was also assessed for a shedding phenotype defined as ≥10 ISH-positive detached cells within the bronchiole lumen. 

### 2.6. nCounter and Bioinformatic Analyses

nCounter was run according to manufacturer’s protocol and custom probe set (Appendix A). RCC output files were assessed for technical QC flags using the nSolver software (nSolver Analysis Software 4.0.70, NanoString Technologies Inc, Seattle, WA, USA) provided by NanoString. Raw reads were next collated from RCC output files into a single count matrix in Rstudio. Because our probe set was highly non-random by design, the number of observed counts for each gene probe were several orders of magnitude greater between mock animals and those acutely infected with SARS-CoV-2. As a result of these large differences in library size, we developed a custom approach for analysis of these data that avoided artifacts introduced by distribution-based normalization methods; our methods are conceptually related to the family of log-ratio transformations that address the compositional nature of sequence-based data. 

To perform these analyses, we leveraged count data within each sample vector as an internal reference, against which we could compare all genes of interest. To do so, we first measured the coefficient of variation (CV) for all genes in the probe set, which included: genes of interest (GOI), housekeeping (HK) genes, positive controls (PCs) and negative controls (NCs). Reassuringly, positive control probes were the least variable of all genes assessed, which was expected given their role in measuring technical variation of the sample prep and assay itself, both of which are demonstrated to be low. We then used the calculated CVs to select the 3 housekeeping genes with the lowest variability. To obtain our within-sample reference value, we calculated the geometric mean of the 3 selected HK genes. We then “tethered” all count data from each sample to its within-sample reference, which we refer to as tethered gene counts (TGCs). 

To visualize data in PCA space and calculate vectors for biplot analyses, we utilized the log2 value of the TGCs. To assess changes in individual genes across groups, we simply used the TGC values. In order to calculate log2 fold-change, we formed a ratio using the TGC for each GOI against its age-matched values from mock animals, and subsequently took the log2 of these values (Figure 1).

To calculate statistical significance, we first checked for normalcy of TGC within each gene/population to be tested using a Shapiro Test. Following BH correction for multiple tests, we observed all data to be normal. We subsequently performed 2-way ANOVA using age and days-post infection as variables of interest. These data were then assessed by Tukey’s post hoc correction, and thus, all reported *p*-values are Tukey adjusted.

## 3. Results

### 3.1. Virologic and Histologic Analysis of SARS-CoV-2 Infection in C57BL/6 Mice

To characterize SARS-CoV-2 infection in C57BL/6 mice, we challenging mice with a previously described variant, SARS-CoV-2-MA10 (MA10) [3]. MA10 was generated from the ancestral Wuhan isolate of SARS-CoV-2 and contains two amino acid mutations within the receptor binding domain of the Spike protein (Q498Y/P499T) that permit binding of Spike to murine Ace2. This double mutant was then serially passaged in Balb/c mice ten times to generate MA10, which contains five additional mutations in NSP4 (T285I), NSP7 (K2R, E23R), Spike (Q493K), and ORF6 (F7S) and results in acute pulmonary disease in infected mice. Intranasal infection with the MA10 strain in Balb/c mice produces an acute and self-limiting pneumonia that was more severe in older animals and less severe in C57BL/6 mice [1]. To verify that MA10 could productively infect C57BL/6 mice, we challenged 10 wk-old male mice intranasally with 10^3^, 10^4^, or 10^5^ PFU MA10 or saline as a vehicle-only control. Mice were then monitored daily for changes in weight. As shown in Figure 1A, while mice challenged with 10^3^ PFU MA10 virus did not lose weight, those challenged with 10^4^ or 10^5^ PFU MA10 exhibited ~15% weight loss 3 days post infection (dpi) and recovered by 7 dpi. It is noteworthy that mice infected with 10^4^ or 10^5^ PFU MA10 experienced a near identical weight loss/recovery trend, suggesting that mice can tolerate a wide range of infectious doses with similar outcomes. 

We also collected tissue from infected mice and mock-infected (mock) controls for histology, RNA-in situ hybridization (RNA-ISH), and qPCR-based measurements of viral load. 10 wk-old C57BL/6 male mice were infected with 10^4^ PFU MA10 or saline. Tissues were collected on days 2, 4, and 6 post-challenge with mock-infected controls being collected on day 6 post-challenge. The experiment was performed with biological tiplicates for each timepoint cohort except for the mock control for which we only had a single mouse. In addition to the lungs, we collected heart, kidneys, and intestines, as these express high levels of Ace2 and are common sites of pathology in patients with severe COVID. The intestinal tract was separated into stomach, duodenum, jejunum, ileum, cecum, and large intestines to determine if there was a preference for SARS-CoV-2 infection across gut tissues. We also collected liver and spleen. We quantified viral RNA by RT-qPCR for the nucleocapsid gene, N2, and for Orf1 (Figure 1B). The highest level of viral RNA was detected in lung on day 2 post infection at 1.2 × 10^8^ copies N2 and 6.9 × 10^6^ copies Orf1 per μL RNA, with the viral RNA level dropping by several logs over the next 4 days. N2 RNA was also consistently detected at day 2 post infection in the stomach and cecum, perhaps as a result from inoculum being swallowed during intranasal infection. N2 RNA was detected inconsistently in the heart, spleen, and large intestines at day 2 post infection and from the spleen at day 4 post infection. Our inability to consistently detect Orf1 RNA in non-lung tissues is likely due to its lower abundance compared to N2 RNA such that Orf1 RNA likely falls below the limits of detection in our assay. The raw data for each mouse at each timepoint are presented in Appendix A. 

To determine the localization of SARS-CoV-2 within the mouse lungs during the course of infection, we performed RNA-in situ hybridization (ISH) for the Spike-coding region, which detects intact virus, defective virions, free viral genome, and subgenomic fragments. At the tissue level, intense bronchial epithelial and wide spread alveolar staining were observed at day 2 post-infection (Figure 1C), a pattern observed in humans but not Balb/c mice [3,25]. By day 4, bronchial epithelial cell ISH staining had largely cleared and there was reduced ISH staining of cells in the airspaces. Finally, by day 6, only scattered cells in the airspaces were ISH-positive. At the cellular level, ISH-positive cells displayed intense staining of their entire cell body and also more focal cytoplasmic staining, possibly representing cytoplasmic virus-containing vacuoles. These intracellular staining patterns are also consistent with what has been observed in cultured cells infected with SARS-CoV-2 [26].

Within the tracheal and main bronchial epithelium at 2 days post-infection, individual infected cells were often identified adjacent to virus-negative neighbor cell, and in some bronchioles, a confluent sheet of infected epithelial cells were observed as abruptly transitioning to a sheet of uninfected cells (Appendix A). Moreover, numerous virus-positive dying cells were detected in the bronchial lumen on day 2 post infection but largely absent at 4 days post infection, suggesting that infected bronchial cells may be shed into the lumen and indicating a mechanism by which infected cells may be eliminated. We also observed intense apical ISH staining of some airway epithelial cells without cytoplasmic labeling, which could represent surface adhesion of the virus without productive infection. Unlike the large swaths of spike-positive cells within the bronchi, signals in alveoli were often limited to individual cells, likely type II pneumocytes as has been reported previously [3]. Outside of the lung, SARS-CoV-2 virus was occasionally detected in some mediastinal lymph nodes and focally in the spleen of one animal. It was not detected in the heart, kidney, liver, or esophagus (Appendix A).

To specifically determine sites of virus replication, we marked replicating virus using an RNA-ISH with a probe that detects the antisense transcript (template strand) of the spike gene (Appendix A). The observed antisense signal was significantly weaker than that of the sense strand, reflecting the relative abundance of sense and antisense strands during infection [26]. As opposed to the diffuse cell body staining that was seen with the sense strand probe, the antisense signal was restricted to discrete foci within the cell, likely representing vacuolized virus-like particles observed by EM (Appendix A) that concentrate transcriptional machinery for viral replication [27] and shield the replicating virus from host antiviral sensors within the cytosol [28]. Consistent with an acute infection followed by viral clearance and recovery, replicating virus was only seen in the lung at day 2 post-infection and was not detected in lungs of 10 wk-old animals at day 4 or 6 post infection.

We also conducted a pathologic examination of all tissues collected from 10 wk-old C57BL/6 male mice infected with SARS-CoV-2-MA10 (Figure 1D). As detected by RNA-ISH analysis, we observed frank cell death within the bronchial epithelium as well as cells that were shed into the bronchial lumen, a phenomenon that peaked on day 2 post infection and significantly decreased thereafter. Peak viral infection at day 2 also associated with prominent endovascular inflammation, with leukocytes attaching to and underlying the endothelium in numerous arterial and venous vessels. However, this marked early endovascular inflammation was not associated with direct viral infection of the blood vessels as determined by RNA-ISH (Figure 1C and Appendix A). As disease progressed, both perivascular and peribronchiolar inflammatory cell infiltrates increased along with alveolar inflammatory infiltrates and pulmonary edema (Figure 1D).

Together, these findings suggest that MA10-C57BL/6 model accurately captures the pathology, tropism, immune cell recruitment to lungs, and disease kinetics that are typical of humans with an acute infection from SARS-CoV-2 [25].

### 3.2. The MA10-R2G Variant

During subsequent expansion of MA10 in Vero cells, we isolated a variant with an R2G substitution in NSP7 (MA10-R2G, R2G henceforth) (Figure 2A). Similar to MA10, infection of C57BL/6 mice with R2G caused maximum weight loss 3 days post infection with a recovery to normal weight by day 6 or 7 post infection (Figure 2B,C). We found that infections with R2G produced a more consistent pattern of weight loss, with lower variability and reduced mortality when compared to MA10 (Figure 2D,E). The improved consistency and reduced virulence of R2G in 10-wk old mice is advantageous in studying potentiating effects of viral infection in older animals and, moving forward, in genetic knockout lines. Therefore, we used the R2G variant in the remainder of the experiments in this study.

### 3.3. Age- and Sex-Dependent Differences in Weight Loss and Survival

In humans with COVID-19, older individuals and males in particular are more likely to develop severe and lethal disease [29,30,31]. To determine the age and sex dependence for SARS-CoV-2 infection of C57BL/6J mice, we infected both male and female young (10 wks), mature (20 wks), and elderly (2 yrs) C57BL6/J mice with 10^3^, 10^4^, or 10^5^ PFU R2G and monitored them daily for changes in weight (Figure 3). Infection of 10 wk-old animals with 10^3^ PFU resulted in no weight loss, while infection with 10^4^ or 10^5^ PFU resulted in 10% weight loss. At all infection doses, the 10 wk-old animals experienced rapid recovery (Figure 3A). In contrast, weight loss in 20 wk-old animals was increasingly severe for each dose compared to 10 wk-old animals (Figure 3B), including significant weight loss in aged mice infected with the lowest dose of 10^3^ PFU. Moreover, there was a clear separation in weight loss curves for animals infected with 10^4^ and 10^5^ PFU, where animals infected with 10^4^ PFU had an average weight loss of 20% and those infected with 10^5^ PFU experienced an even greater degree of weight loss. Importantly, while none of the 10 wk-old mice infected with 10^5^ PFU succumbed to infection, 40% of the 20 wk-old animals infected with 10^5^ PFU succumbed to infection or reached the euthanasia criteria of 30% weight loss.

We predicted that the 2 yr-old animals would be extremely sensitive to infection.

Therefore, we initially dropped the dose range and infected 2-yr old mice with 10^2^, 10^3^, or 10^4^ PFU (Appendix A). Indeed, 2 yr-old males infected with as low as 10^2^ PFU did not fully recovery and ended the 7-day study course with an average of 10% weight loss, while those infected with 10^3^ PFU exhibited an average of 25% weight loss, confirming increased sensitivity to SARS-CoV-2 infection at lower virus challenge doses. Moreover, all males infected with 10^4^ PFU succumbed to infection or reached the euthanasia criteria of 30% weight loss, occurring within day 5 post-infection. In contrast, 2 yr-old female mice responded similarly to 20 wk-old females infected with 10^3^ or 10^4^ PFU but surprisingly without mortality. We repeated this study but increased the virus challenge dose range to 10^3^, 10^4^, or 10^5^ PFU (Figure 3C). As in our prior experiment, female, 2 yr-old mice responded similarly if not better in terms of weight loss than 20 wk-old mice at all virus challenge doses, and strikingly, none of the 2 yr-old females succumbed to infection, suggesting that older female C57BL6/J mice are less sensitive to infection than 20 wk-old female mice. A similar result was reported for influenza infection in female mice [32]. As in humans, older male mice fared the worst following SARS-CoV-2 infection, with no mice surviving past day 5 post infection with 10^4^ PFU R2G. Together, these data demonstrate a clear age and sex-dependent difference in C57BL/6 mice in response to infection that mimics the epidemiology of COVID-19 in humans. 

### 3.4. Age- and Sex-Dependent Changes in Virus Persistence and Localization

To determine how age and sex affect the ability of mice to clear virus and recover from infection, 10 wk-, 20 wk-, and ~2 yr-old mice were infected with 10^4^ PFU R2G; and lung, heart, kidney, spleen, liver, stomach, and duodenum were collected 2, 4, and 7 days post infection with tissues from mock-infected animals being collected on day 7, each including triplicate samples across cohorts. Viral load per tissue and time point as well as weight loss for each cohort are presented in Figure 4. Of note, because a significant number of 2 yr-old male mice succumbed to infection or met euthanasia criteria on day 5 post infection, the data for males on day 7 post-infection represent only the survivors. As such, data for day 7 males suffer from selection bias and likely under-represent the true extent of disease severity in older male mice.

The highest viral loads in 10 wk-old mice infected with 10^4^ PFU R2G were found in the lungs. At day 2 post infection, males had a higher viral load (5.7 × 10^9^ copies of N2 RNA per mg lung tissue) compared to females (2.5 × 10^9^ copies of N2 RNA per mg lung tissue). Subsequently, viral load dropped by four logs to 3.2 × 10^5^ and 4.9 × 10^5^ copies N2 RNA per mg lung tissue in males and females, respectively, by 7 dpi. Virus was also detected in the heart and stomach of males and females and in the spleens of females only at 2 dpi but not thereafter.

For 20 wk-old animals (Figure 4B), we also detected the highest viral copy number in the lungs, with 7.1 × 10^9^ and 4.6 × 10^9^ viral copies per mg lung tissue in males and females, respectively, 2 dpi and dropping several logs by 7 dpi to 1.1 × 10^7^ and 4.8 × 10^6^ copies N2 RNA per mg tissue in males and females, respectively. While there was no significant difference in viral load when comparing male 20 wk-old animals to age-matched females (Figure 4E), their average viral load at day 7 post infection was 19-fold higher (*p* = 0.0015, unpaired t test) than levels in 10 wk-old animals. Additionally, virus in 20 wk-old animals was detected in additional tissues and at later time points in comparison to 10 wk-old animals (Figure 4 and Appendix A), indicating that older animals clear virus less effectively (Figure 4D).

When males and females were averaged in 2 yr-old mice, the pattern of lung viral load essentially reproduced that observed in the lungs of 20 wk-old mice (Figure 4B–D). However, while 20 wk-old male and female mice had nearly identical viral RNA levels at each time point, the abundance of lung viral RNA in the 2 yr-old male and female animals diverged, with males having more virus and females having less virus than the 20 wk-old animals at each time point (Figure 4E). Additionally, quantitation of viral RNA abundance demonstrated an increase in the distribution and durability of virus in tissues from animals infected at ~2 yr of age, with virus detected in every tissue assayed (see Figure 4 and Appendix A). These observations are consistent with the sex- and age-dependent differences in weight loss and survival (see Figure 3). Moreover, with higher viral loads at later time points and a greater distribution of virus-positive tissues, these data support the notion that, like humans, older mice are unable to efficiently clear virus. 

### 3.5. RNA-ISH and Pathology of R2G Infection in Young, Mature, and Elderly Mice

Next, we assessed the distribution of virus within the lungs of R2G-infected young, mature, and elderly mice by RNA-ISH (Figure 5). ISG staining patterns for younger mice infected with R2G largely phenocopied that observed for mice infected with MA10. RNA-ISH for the Spike gene was most intense in bronchial epithelial cells 2 days post infection with widespread airspace involvement. We also observed virus-positive cells sloughed into the lumen of the bronchioles. The ISH staining distribution was greatly diminished on days 4 and 7 compared to day 2 post infection. However, relative to 10 wk-old animals infected with R2G, there was more widespread and persistent RNA-ISH signal in 20 wk-old and 2 yr-old animals at 4 and 7 dpi. Interestingly, 20 wk-old and 2 yr old male mice had more widespread RNA-ISH staining at day 7 compared to their female counterparts. Together, these data indicate that SARS-CoV-2 increasingly spreads within lungs of aged mice compared to young mice and more so in aged males. 

We next conducted a pathologic examination of H&E-stained sections from mice infected with R2G (Figure 6 and Appendix A). As noted above, the true extent of disease severity in 2 yr-old males is likely underestimated at day 7 due to the loss of many of the infected males at day 5 post infection. Despite this dynamic in the aged male cohorts, infection with R2G led to a significant number of dead/dying cells within the bronchi of older mice. While there was no difference in the degree of cell death as animals aged, the severity of this pathology varied more in the older compared to young animals (Figure 6A). We also noted a pronounced burst of endovascular inflammation on day 2 post-infection for all ages, (Figure 6B,C) which tapered off through day 7 post-infection. Concurrent with the decrease in endovascular inflammation, there was a marked increase in perivascular and peribronchial inflammation (Figure 6D–F), indicating that inflammatory cells were moving out of the vasculature and into the tissue during the course of the infection. While there were generally no age-dependent changes in vascular and bronchial inflammation scores due to infection, perivascular inflammation was significantly higher in 2 yr-old mock-infected mice than younger mice (Figure 6E,F), suggesting a greater degree of basal perivascular inflammatory cells in older animals, which may represent increased bronchus associated lymphoid tissue (BALT) in aged mice [33]). Despite the lack of apparent differences in inflammatory responses across ages, we identified a significant increase in interstitial pneumonia and pulmonary edema that was both infection- and age-dependent (Figure 6G,H). Interestingly, in contrast to infection with MA10, infection with R2G did not cause edema in 10 wk-old animals, likely reflecting its attenuated virulence of this viral variant. Older mice infected with R2G, however, demonstrated significant edema and older mice in general also had more histopathologic evidence of comorbidities, including amyloidosis, lymphoid hyperplasia/lymphoma, pulmonary adenomas, and hyaline glomerulopathy (Appendix A).

### 3.6. Dependence on Innate Immune Response for Disease Recovery and Early Reductions in Viral Load

To evaluate the roles of innate and adaptive immunity in control of SARS-CoV-2 and recovery from disease, we compared wild type and SCID mice (Figure 7), as SCID mice lack an adaptive immune response due to the absence of functional B- and T-cells but have an intact innate immune response for defense against virus infection. Male SCID mice were challenged with 10^3^, 10^4^, and 10^5^ PFU R2G and compared to 10 wk-old wild type C57BL/6 mice challenged with the same doses. Similar to wild type mice and for all infection inoculums tested, SCID mice experienced maximal weight loss at day 3 post infection, began to recover on day 4, and reached their pre-infection weight on day 6 (Figure 7A). These results indicated that the adaptive B- and T-cell-mediated immune responses are dispensable for the ability to recover from acute SARS-CoV-2-disease as marked by weight loss.

To determine whether adaptive immune response is necessary for the clearance of SARS-CoV-2 or pathologies resulting from infection, we challenged SCID mice with 10^4^ PFU and collected tissue 2, 4, or 7 dpi. Viral load on days 2 and 4 post infection were indistinguishable from wild type animals. However, despite the ability of SCID mice to return to normal weight by day 7 post infection, SCID mice had an elevated viral load 7 days post infection compared to wild type animals (4.8 × 10^7^ and 3.2 × 10^5^, *p* = 0.04, 1-tailed *t*-test, respectively) (Figure 7B), indicating that adaptive immunity is necessary for clearance of virus at later infection time points. Furthermore, 10 wk-old SCID mice infected with R2G demonstrated a similar trend in tissue pathology compared to wild type 10 wk-old mice (Figure 7C and Appendix A), with the most notable differences being increased endovascular inflammation at day 4 and decreased interstitial and peribronchovascular inflammation at day 7. These observations indicate that control of viral burden is a multiphasic process that is initiated by innate immune responses and that transitions to adaptive responses.

### 3.7. Innate Immune Activation

To determine if innate immune activation is triggered in response to SARS-CoV-2 infection in the C57BL/6 model, we interrogated innate immune gene expression using a custom NanoString innate immune activation and response gene mouse probe set and nCounter-based transcriptional analyses of the lung samples described above (Figure 8). The probe set (which included assay controls and housekeeping genes (Appendix A)) was designed to monitor innate immune activation through expression of (1) IRF3-target genes, (2) types 1 and III interferon (IFN), (3) interferon-stimulated genes (ISGs) that are known to respond to specific pathogen recognition receptors, and (4) select NFkB-responsive genes encoding inflammatory response mediators. Resulting mRNA counts were transformed to account for differences in library size and then assessed by principle component analysis (PCA) (Figure 8A–D and Appendix A). Sample separation across the first principal component was predominantly driven by the acute induction (upregulation) of specific genes in the panel following infection (Appendix A); these included known IRF3-target genes Rsad (viroporin), Cxcl10, IFN-b, Isg15, Ifi44, and Ifit2 [34] and ISGs including Mx1, Mx2, Isg20, Oas1b, and others; parallel induction of Tnfa, IL6 and IL-1 was also detected. Gene expression across lung samples along this first principal component largely clustered by time point, with samples from 2 days post infection demonstrating the highest levels of innate immune gene induction; samples at day 4 and 7 shifted toward mock levels (Figure 8A). Moreover, as shown in Figure 8B, the spread of samples from left to right (PC1) also strongly correlated with viral load (presented in Figure 4), suggesting that variation in expression levels of the innate immune genes measured here was primarily driven by viral burden. While PC2 explained far less of the total variation in the data set (6% vs. 74.9%, PC2 vs. PC1, respectively), we identified a major distinction along this axis that separated the 10 and 20 wk-old animals from the 2 yr-old mice (Figure 8C) due to differences in IL-18, Tlr4, Irf2, IL1a, IfngR1, Irf7, and IfnaR1 (Appendix A). 

Given the clear delineation between the oldest and youngest mice across the data set, we next considered the degree to which innate immune profiles differed at baseline. We note that based on pathology scores (discussed in Figure 6), 2 yr-old mice had increased levels of inflammation at baseline. Comparison of gene expression levels suggested that these differences are associated with the elevated baseline levels of Tlr7 (*p* = 0.0022), TnfRsf1B (*p* = 0.0022), Gm14446 (*p* = 0.0022), Irf7 (*p* = 0.0043), Mx1 (*p* = 0.0043), Cxcl10 (*p* = 0.008), Dhx58 (*p* = 0.008), Tnfa (*p* = 0.015), Ifih1 (*p* = 0.026), Rsad2 (*p* = 0.026), Ccl12 (*p* = 0.041), Isg15 (*p* = 0.041), Ifit2 (*p* = 0.041), Eeif2ak (*p* = 0.041) (Appendix A). To assess the degree to which these elevated levels of gene expression altered the output of the innate immune response in 2 yr old mice, we calculated the log2-fold change for each gene by comparing expression levels at each time point with age-matched mock-infected animals (Figure 8F). Our analyses indicated that despite the presence of baseline inflammation, 2 yr old mice had significantly lower induction of innate immune genes at 2 days post-infection, including Il6, Ccl4, Cxcl10, Rig-I, Irf7, Ifna2, Oas1b, and Mx2. As we considered the impact of age more closely, we noted that while 10 wk-old animals clustered tightly at each time point (Compare Figure 8A,C), there was little to no separation between 20-wk old mice at days 4 and 7 post infection. 2 yr-old animals followed this pattern, with even less separation between days 4 and 7 post infection. These observations show that the resolution of the innate immune response was delayed in older animals. Indeed, several key antiviral effectors and inflammatory drivers remained upregulated at late infection time points in older mice when compared to the 10 w-old animals, including several that were poorly upregulated initially in the older cohort; these included Il6, Isg15, Cxcl10, and Ifit2. Taken together, our transcriptional analyses (1) indicated baseline inflammation was present in older mice, in agreement with pathology scores; (2) demonstrated key deficits in the acute innate immune responses in older mice; and (3) revealed incomplete resolution of the innate immune and inflammatory responses in older mice at 7 days post infection, when these responses in younger mice had already returned to baseline levels. 

## 4. Discussion

We provide a model of SARS-CoV-2 infection and recovery in C57BL/6 mice, featuring the mouse-adapted R2G virus strain which is a variant of the MA10 strain. We present a robust and comprehensive comparison of early to late infection in young, mature, and elderly adult mice, both in males and females. Our analyses include assessments of virus density and localization, animal weight loss kinetics and mortality, disease pathology across multiple tissues, and innate immune responses. We show that as in humans [29,30,31], disease is mild and quickly resolved in young, healthy individuals; however, disease severity is age- and sex-dependent, with older animals (especially older males) showing greater sensitivity to lower virus challenge doses, increased mortality, increased virus persistence with systemic virus spread, and greater lung damage. We show that the magnitude of the innate immune activation displayed by innate immune gene induction is greatest in young mice and that virus and the innate immune/inflammatory responses persist longer in aged mice, suggesting that these altered responses compared to young mice contribute to viral persistence and increased lung injury in elderly mice. Finally, we show through comparison of wild type and SCID mice that the innate immune response is the primary driver of weight recovery and early stage viral control in vivo, while an intact adaptive response is still required for efficient virus clearance at later infection time points. Together, these data characterize the effects of SARS-CoV-2-R2G infection in C57BL/6 mice, which are remarkably similar to those in humans [25,29,30,31], as well as what was seen for the 2003 strain of SARS1 [35,36]. Additionally, this study paves the way for future studies aimed at identifying key immune pathways that are required for recovery and disease pathology following infection with not only the original strain of SARS-CoV-2 but future emergent coronaviruses as well.

A number of excellent groups have previously reported model systems for the in vivo study of SARS-CoV-2 infection in mice. Early in the pandemic, these studies required the use of mice expressing human Ace2 since original isolates of SARS-CoV-2 did not infect mice due to differences in human and murine Ace2 [37,38]. While informative, infections in these mice did not replicate human disease due to the inappropriate localization and expression of hAce2. As the pandemic progressed, several groups reported on mouse-adapted strains of SARS-CoV-2 that acutely infected mice and produced disease reflecting what was seen in humans at that time [3,6,7,8,9,10,11,12,13,14]. Though several of these studies examined the effects of infection in C57BL6 mice [3,7,10,11,12] these were limited to single sex or age cohorts or employed strains of SARS-CoV-2 that were too virulent for elucidating immune mechanisms of recovery by genetic screens. Compared to the few studies in C57BL/6 mice, the most comprehensive early studies with mouse-adapted strains characterized infection in Balb/c mice [3,7,11,13] which do not permit the vast number of genetic screens that are possible with transgenic and gene knockout lines on the C57BL/6 background. Important to our future goals of genetically dissecting key immune pathways for disease recovery of knockout mouse strains on the C57BL/6 background, the immune responses and disease severity observed in Balb/c and C57BL/6 mice are different. For example, Balb/c mice are skewed towards a Type II immune response, which may exacerbate disease as in severe COVID, while C57BL/6 mice are skewed toward a Type I antiviral/inflammatory immune response, which is needed for efficient clearance of virus [39]. This notion is supported by studies showing that Balb/c mice experience more severe disease compared to C57BL/6 mice of the same age [3]. This and other differences necessitated the more comprehensive study of SARS-CoV-2 infection in C57BL/6 mice presented here.

Interestingly, in addition to their usefulness in genetic screens, C57BL/6 mice may better reflect disease recovery in humans than Balb/c mice. Specifically, virus tropism in C57BL/6 lungs appears to more closely phenocopy that in humans, where intense staining was observed in bronchial epithelium in addition to the alveolar space. In contrast, SARS-CoV-2 does not apparently localize to the bronchial epithelium in Balb/c mice ([3,25] and present study). Moreover, staining of bronchial epithelium in humans and C57BL/6 mice correlates with shedding of dying cells into the bronchial lumen ([3,25] and present study). Whether this outcome is important in virus clearance or virus spread or in the increased disease pathology in Balb/c mice relative to C57BL/6 mice is unclear. While not necessarily exclusive to C57BL/6 mice, the prominent pulmonary edema and interstitial pneumonitis seen in the older mice in our study also corresponds with the pulmonary edema seen radiographically in humans, and suggests that this phase of injury may resemble the early stage of SARS-CoV-2 induced acute respiratory distress in humans underlying COVID-19 [40,41]. Our finding that infection-mediated lung injury follows clearing of the virus, and that more extensive lung injury is observed in older mice that have delayed viral clearance, suggests that a robust innate immune response in young mice leads to rapid clearing of the virus and limits the immunopathology in the lung, processes that ascertain SARS-CoV-2 infection outcome in most humans. Together, these observations suggest that C57BL/6 mice accurately capture infection and outcome dynamics observed in humans.

Novel variants of SARS-CoV-2 have arisen faster than detailed studies aiming to characterize their properties can be completed before being replaced by a new emergent variant. In fact, more recent variants, including the current omicron variants, developed the ability to infect mice and rats [19,20,21], which share Ace2 sequence similarity in the residues to which SARS-CoV-2 binds. What drove this adaptation is uncertain, but it is interesting that cryptic SARS-CoV-2 variants were found in the sewers of New York City that contain mutations that are found in mouse adapted MA10 and R2G as well as in omicron [42]. However, while omicron is able to infect mice, it is important to note that omicron-induced disease is much different from that driven by the ancestral SARS-COV-2 and appears to preferentially infect upper airways as opposed to lower airways and with milder symptoms, both in humans and mice [19,20,21,43]. Moreover, while the ability of omicron to infect mice provides the very important opportunity to study natural infections of current strains in vivo, this variant does not allow studies of recovery from the more severe infections and disease pathologies engaging the lower airway that better reflect those arising from the ancestral virus and earlier pandemic variants. 

As with many viral infections in humans, older individuals and older males in particular are more sensitive to infection with SARS-CoV-2 and are more prone to experience and succumb to severe disease [22,23,24,44]. This is likely due to a combination of factors that include comorbidities such as heart, lung, renal, and liver disease, obesity, dementia, and cancer which are more prevalent in older individuals [45] as well as X-lined expression of many immune genes and the contrary responsiveness of immune cells to estrogen/progesterone and testosterone [22]. We show here, though a comprehensive analysis of the effects of both age and sex on SARS-CoV-2 infection in mice, that older mice are much more sensitive and susceptible to infection than younger mice (10-wk), consistent with what is seen in humans. Surprisingly, though elderly (2 yr) female mice are less susceptible to infection than mature adult (20 wk) female mice. The cause of this difference is uncertain but held true over multiple independent experiments within this study; however, a similar outcome was also reported for influenza A virus infection in mice [32]. In humans, increased levels of inflammatory cytokines such as IL-6 were found in older males with severe COVID whereas increased levels of anti-inflammatory cytokines like IL-10 were observed in older females 14 days post infection than in males [23]. Likewise, we observed increased levels of IL-6 in elderly mice by NanoString analysis innate immune gene expression, but our study did not have the power to segregate males from females, necessitating future studies to elucidate the molecular and immunologic underpinnings driving the sex-based differences to SARS-CoV-2. Importantly, these observations further validate this model system as a relevant model for the study of SARS-CoV-2-mediated disease and recovery.

In conclusion, we present a comprehensive analysis of SARS-CoV-2 infection and recovery in C57BL/6 mice, showing responses, pathologies, tropism, distribution, and age- and sex-dependent differences in disease severity that mimic what was seen following infection with the original isolates of SARS-CoV-2, when the virus first emerged in humans. These observations serve as a basis from which future C57BL/6-based genetic screens can be compared. Additionally, due to the increased reproducibility of R2G over the parental MA10, infection of C57BL/6 mice with the R2G variant of SARS-CoV-2-MA10 is an ideal model for SARS-CoV-2 mediated disease and recovery. 

## Figures and Tables

**Figure 1 vaccines-11-00047-f001:**
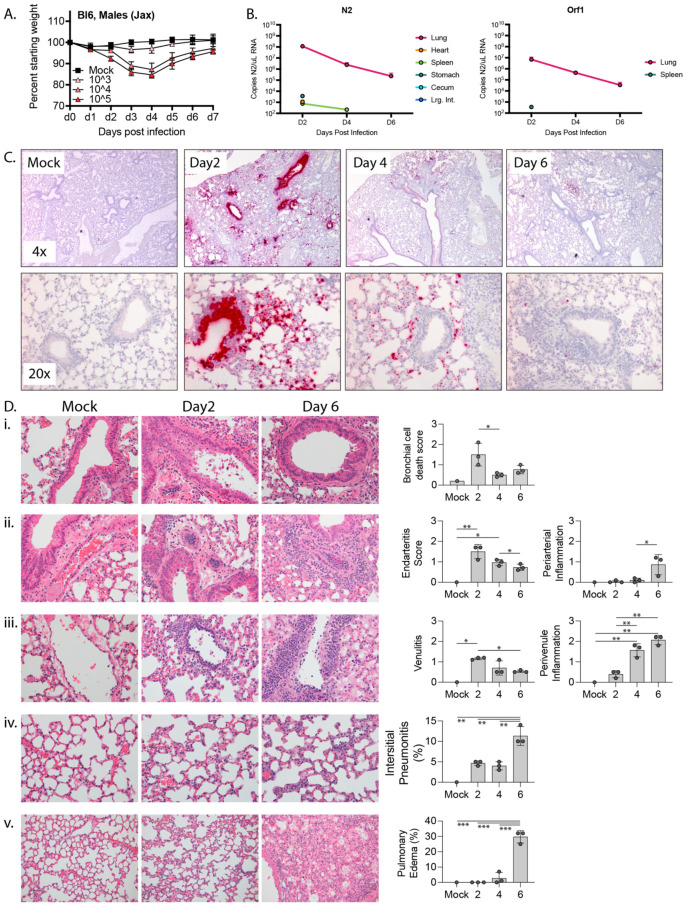
Response to mouse adapted SARS-CoV-2 (MA10) in 10 wk-old C57BL/6 mice. (**A**) 10 wk-old C57BL/6 mice were mock infected or infected with 10^3^, 10^4^, or 10^5^ PFU MA10 via intranasal administration and monitored daily for changes in weight, with 5 male mice per cohort. (**B**) Copies of MA10 per μL RNA as determined by qPCR of N2 and ORF1. (**C**) Representative images at 4x (top) or 20× (bottom) of RNA-ISH (in situ hybridization) against the sense strand of Spike (red) from mock infected lungs or lungs collected 2, 4, or 6 days post infection with 10^4^ PFU MA10. (**D**) H&E-stained lungs and bar graphs showing (i) bronchiolar epithelial cell death (400×), (ii) arterial inflammation (400×), (iii) venous inflammation (400×), (iv) interstitial pneumonitis (400×), and (v) edema (200×) from mock infected or MA10-infected lungs collected 2 or 6 days post infection. Significance calculated by one-way ordinary ANOVA using Tukey’s correction for multiple comparisons and reflected by * (*p* ≤ 0.05), ** (*p* ≤ 0.01), and *** (*p* ≤ 0.001). For B-E there were 3 male mice per cohort except for mock which only had one animal.

**Figure 2 vaccines-11-00047-f002:**
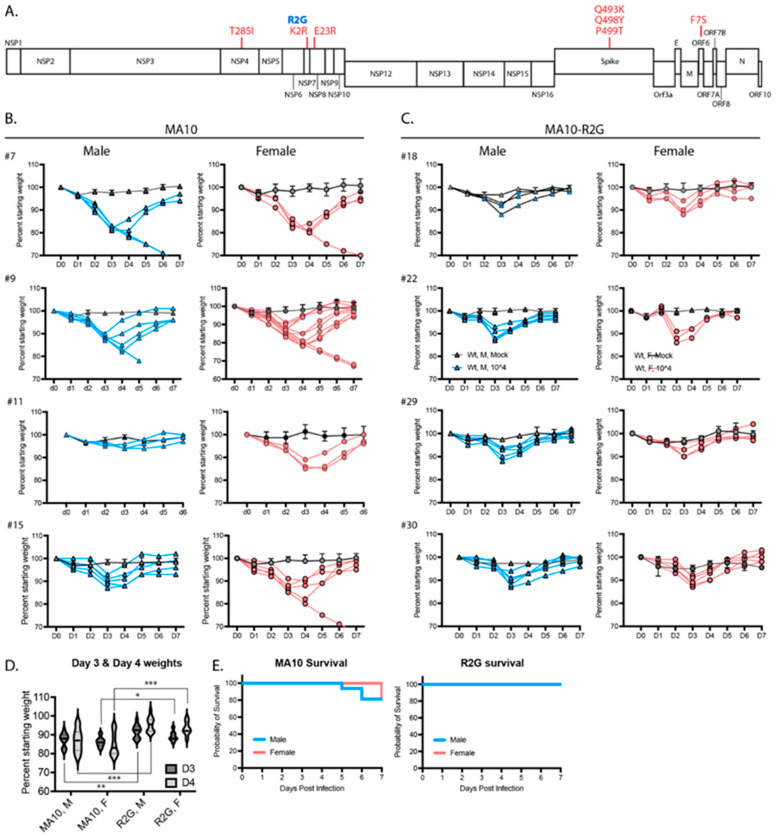
Comparison of SARS-CoV-2 MA10 and the R2G variant that was generated in our lab. (**A**) Schematic representation of the SARS-CoV-2 genome showing mutations found in MA10 (red) and in the R2G variant (blue) as compared to the original Wuhan isolate. (**B**,**C**) Weight loss in males (blue) and females (pink) infected with MA10 (**B**) or R2G (**C**). Four independent experiments are shown for each to demonstrate experiment-to-experiment variability/consistency. Data for mocks are shown as averages with 2–5 mice per cohort, while spaghetti plots are presented for each infected animal. (**D**) Violin plots of body weight for male (M) and female (F) animals three or four days post infection with either MA10 or R2G. Plots include all animals shown in (**B**,**C**) and significance was calculated using Welch’s *t*-test, where * is *p* ≤ 0.05, ** is *p* ≤ 0.01, and *** is *p* ≤ 0.001. (**E**) Survival curves for animals infected with MA10 or R2G as presented in (**B**,**C**).

**Figure 3 vaccines-11-00047-f003:**
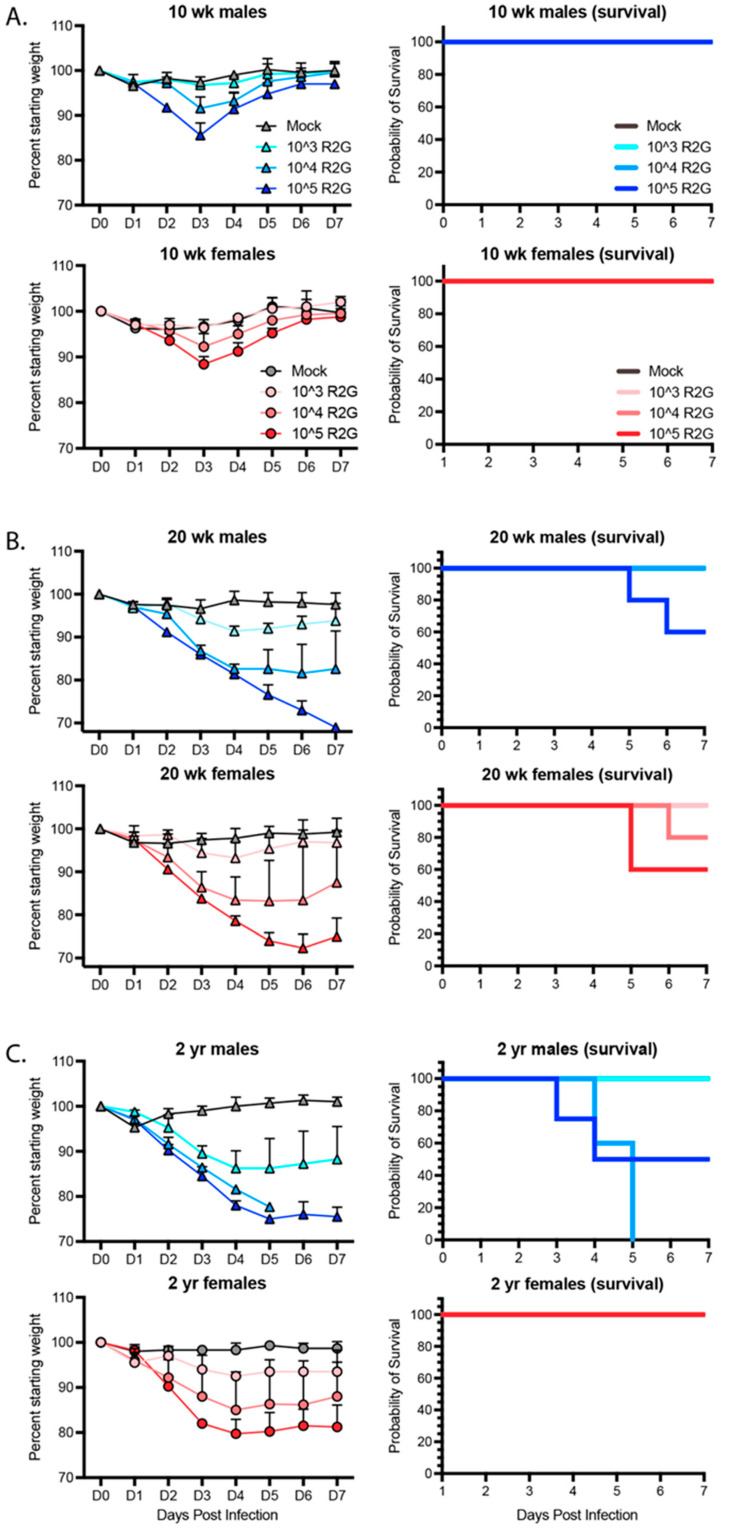
Dose, age, and sex dependent response to R2G in terms of weight loss and survival for 10 wk-old (**A**), 20 wk-old (**B**), and 2 yr-old (**C**) mice infected with 103, 104, or 105 PFU R2G, with 3–5 mice per cohort. Mice were infected with indicated doses (PFU) by intranasal administration and followed daily for weight loss. Mice were euthanized if they fell below 70% original weight or became otherwise moribund. Appendix A shows dose response for 2 yr-old male and female mice treated with 102, 103, or 104 PFU R2G.

**Figure 4 vaccines-11-00047-f004:**
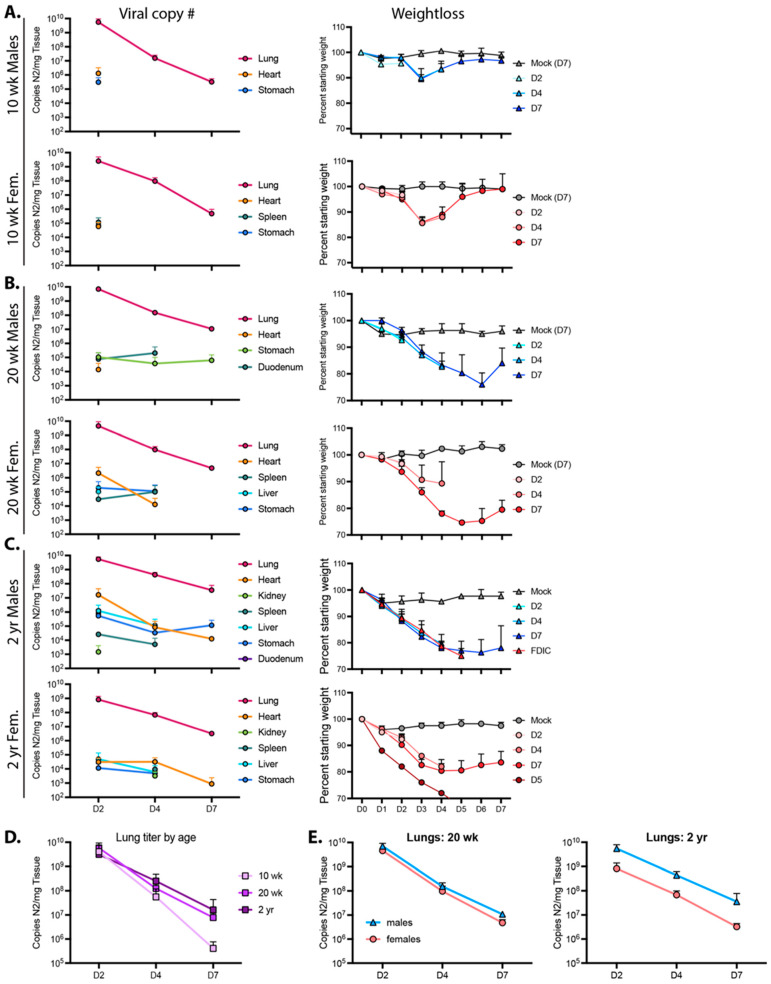
Viral reads and weight loss from 10 wk-old (**A**), 20 wk-old (**B**), and 2 yr-old (**C**) male and female mice infected with the R2G variant. Viral reads were determined by qPCR for copies of Nucleocapsid RNA (N2) per mg tissue, which was collected 2, 4, or 7 days post infection, with 3 mice per cohort. Tissues that are positive for viral reads are graphed for each cohort, and Ct cut off of 34 was determined from standard curves. Of note, while all mice at all time points were positive for virus in the lung, this was not the case for other tissues, where positivity may have been confined to one or two mice with the cohort. These data are presented in Appendix A. Additionally, shown are weight loss for animals used to determine the number of viral read per tissue. Male (blue triangles) and female (pink circles) animals euthanized 2, 4, or 7 days post infection are shown as D2, D4, and D7, respectively. Of note, data from D7, 2 yr-old males may under-represent true disease severity as many 2 yr-old males succumbed to infection or met euthanasia criteria on day 5 post infection and data from D7 2 yr-old males could only be surveyed from survivors. (**D**,**E**) Lung titers from (**A**–**C**) with sexes combined (**D**) or separated (**E**); lung titers were not significantly different between age and sex by 2-way ANOVA.

**Figure 5 vaccines-11-00047-f005:**
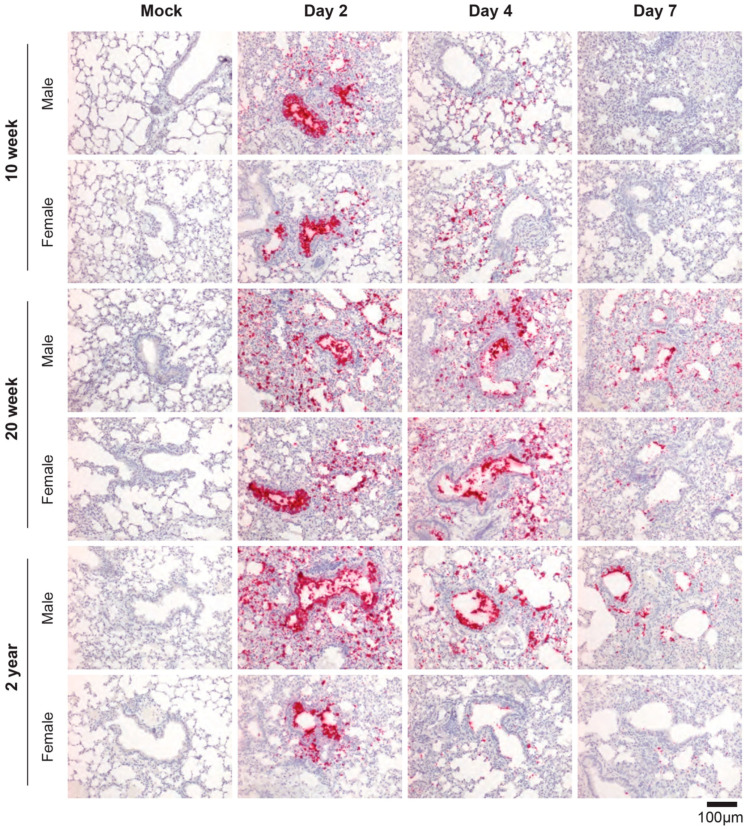
SARS-CoV-2-specific RNA-ISH of 10 wk-old, 20 wk-old, and 2 yr-old mice infected with R2G. Formalin-fixed, paraffin-embedded (FFPE) tissue sections were stained for the sense strand of spike as in Figure 1C. Tissue was collected either 2, 4, or 7 days post infection or 7 days post mock infection from the 10 wk-, 20 wk-, and 2 yr-old male and female mice discussed in Figure 4, and quantitation of air space involvement can be found in Appendix A.

**Figure 6 vaccines-11-00047-f006:**
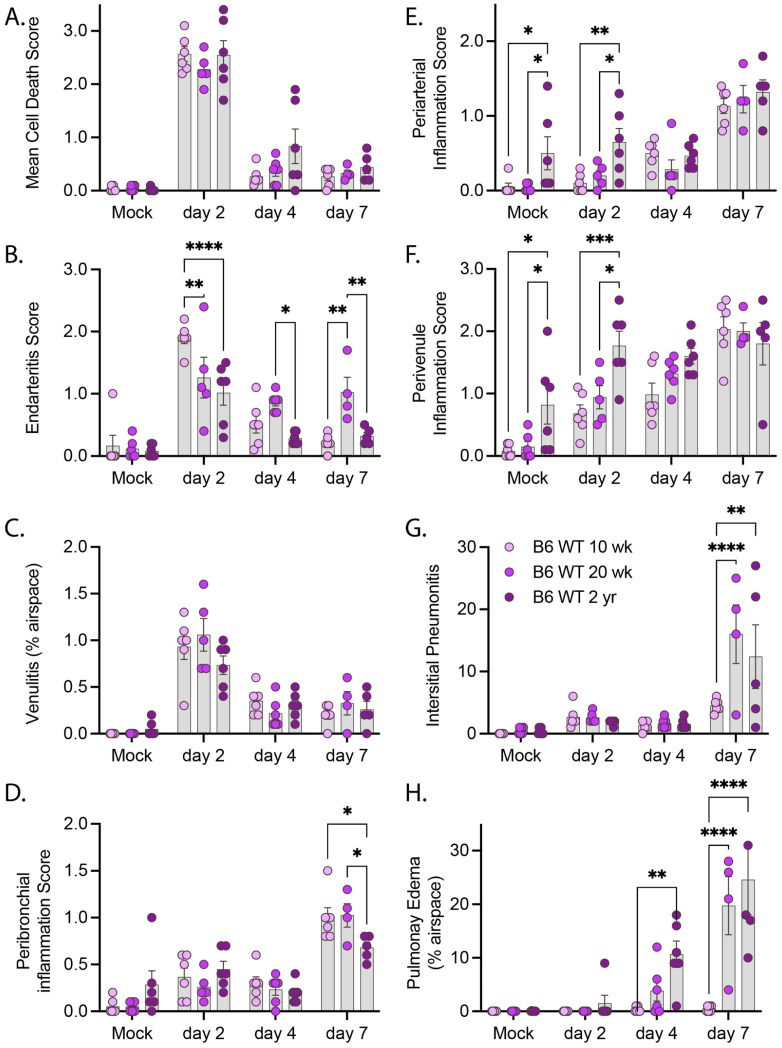
Pathology summary of lung tissue from of 10 wk-old, 20 wk-old, and 2 yr-old mice infected with R2G. Tissue was collected either 2, 4, or 7 days post infection or 7 days post mock infection from the 10 wk-, 20 wk-, and 2 yr-old male and female mice discussed in Figure 4. Data represent pathology scores from H&E stained FFPE tissue sections, and representative images can be found in Appendix A. (**A**) Mean Cell Death, (**B**) Endarteritis, (**C**) Venulitis, (**D**) Peribronchial inflammatory, (**E**) Periarterial inflammation, (**F**) Perivenule inflammation, (**G**) Interstitial Pneumonitis, and (**H**) Pulmonary Edema. Significance tested by 2-way ANOVA with Tukey’s adjustment for multiple comparisons, where * is *p* ≤ 0.05, ** is *p* ≤ 0.01, *** is *p* ≤ 0.001, and **** is *p* ≤ 0.0001.

**Figure 7 vaccines-11-00047-f007:**
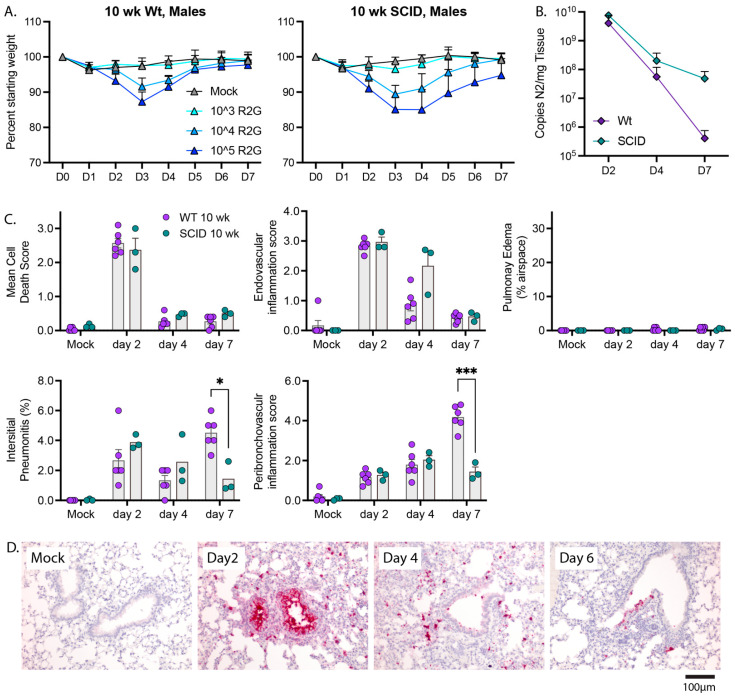
The adaptive immunity is not required for recovery from weight loss but is required for late time point control of virus. (**A**) Comparison of the dose response in regard to weight loss and recovery between 10 wk-old wt and SCID males with 5 mice per cohort. Note, data for wt males are repeated from Figure 3A. (**B**) Comparison of viral copy number between 10 wk-old wt animals and SCID males as determined by qPCR for copies of N2 per mg tissue. Note, data for wt animals are repeated from Figure 5 with 3 mice per cohort (**C**) Comparison of lung pathology between wt and SCID animals presented in B. Significance tested by 2-way ANOVA with Tukey’s adjustment for multiple comparisons, where * is *p* ≤ 0.05, and *** is *p* ≤ 0.001. (**D**) RNA-ISH for Spike sense strand. H&E stained FFPE tissue sections, and representative images can be found in Appendix A.

**Figure 8 vaccines-11-00047-f008:**
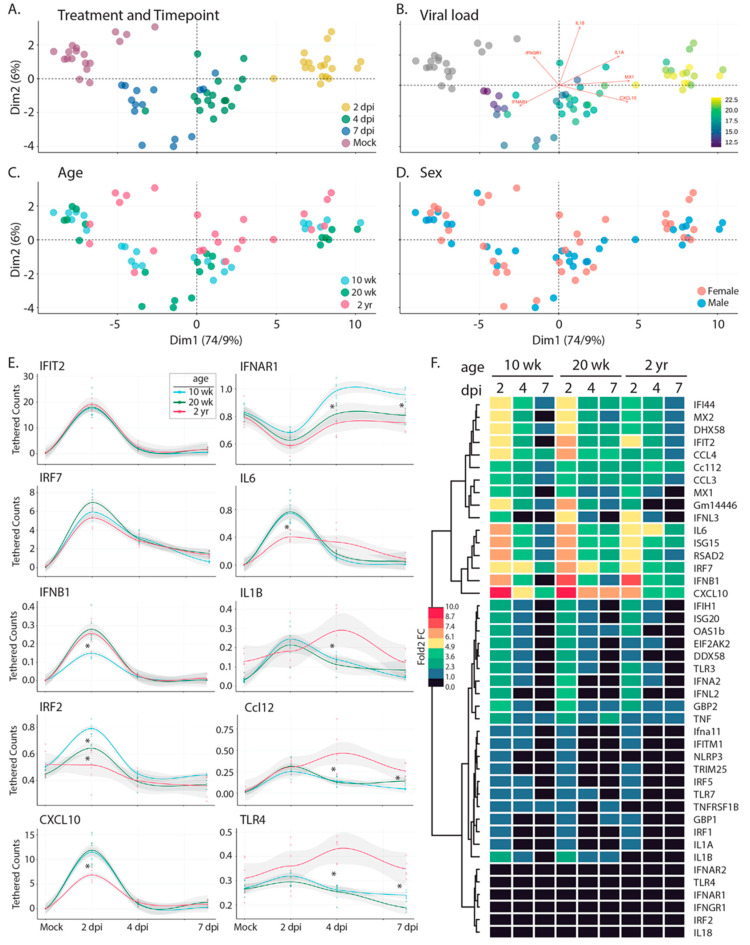
Bioinformatic analysis of the innate immune response following infection with R2G. Tissue was collected either 2, 4, or 7 days post infection or 7 days post mock infection from the 10 wk-, 20 wk-, and 2 yr-old male and female mice (Figure 4). Gene expression of 43 innate immune genes (Appendix A) was determined using the nCounter by NanoString. Raw expression and meta data can be found in Appendix A, respectively. (**A**–**D**) PCA analysis of expression data for all mice overlaid with meta data for treatment and timepoint (**A**), viral load (**B**), age (**C**), and sex (**D**). Scree plots of eigenvalues of factors driving PC1 and PC2 can be found in Appendix A. (**E**) Selected smoothed line plots for single gene expression data from 10 wk-, 20 wk-, and 2 yr-old mice for treatment and timepoint. Lines show smoothed average of tethered counts. Circles show individual animals. Grey shading shows 95% confidence interval. Asterisks mark significant differences as determined by Tukey’s adjustment for multiple comparisons where * is *p* ≤ 0.05 (see Appendix A). Plots for all genes analyzed can be found in Appendix A. (**F**) Heat map showing log 2-fold change gene expression relative to mock treated controls.

## Data Availability

Datasets used for the analyses in this study have either been added to Appendix A or can be requested from the corresponding author.

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
