# Peer review of "A C57BL/6 Mouse Model of SARS-CoV-2 Infection Recapitulates Age- and Sex-Based Differences in Human COVID-19 Disease and Recovery"

_vaccines, 2022, doi:10.3390/vaccines11010047_

Round 1

Reviewer 1 Report

Comments to the author:

The manuscript entitled “A C57BL/6 Mouse model of SARS-CoV-2 infection recapitulates age- and sex-based differences in human COVID-19 disease and recovery”. This study SARS-CoV2 mediated disease in C57BL/6 mice accurately phenocopies human disease across ages and establishes a platform for future therapeutic and genetic screens for not just SARS-CoV-2 but also novel

coronaviruses that have yet to emerge. The study is presented nicely.

Comments

The authors are required to measure protein expression in various tissues

Author Response

Comments to the Reviewer

We appreciate the reviewer’s time and efforts in critiquing our study as well as for their opinion that this was a nicely presented study.  We also politely disagree that protein analyses are needed.  We have provided RNA levels, which are widely accepted as a measure of viral and host gene expression across the field, as the biomarker of gene expression, thus the protein expression data sets would be largely redundant with the gene expression data shown.  Moreover, to conduct protein expression analyses at this point would require the use and death of numerous mice which would not compliant with the ethical use of animals nor supported by IACUC standards.

Reviewer 2 Report

The manuscript submitted by Davis et al. is a study into the ability of a variant of SARS-CoV-2 to infect mice. A prior study uncovered SARS-CoV-2 MA10 was able to infect Balbc mice. The current study tested whether C57 mice could be infected and show signs of Covid-19 following inoculation with SARS-CoV-2 MA10. The study is an advancement of the field, albeit incremental. The submitted investigation found two additional mutations present following serial inoculations.

Strengths

This is a fairly in depth characterization of Covid-19 pathogenesis in mice following inoculation with this virus.

Many aspects of Covid-19 are considered and tested; including age, sex, multiple models, and various organs.

Weaknesses

The limitations should be more detailed to include descriptions of the strengths/weaknesses of the mouse adapted virus. The authors could also describe the advantages/disadvantages for using this virus/mouse to study Covid-19.

Illustrating where these mutations are on the virus and the effect on binding to ACE2 would be helpful.

It would be helpful to better differentiate R2G from SARS-CoV-2 MA10.

A lot of data is presented, however it was difficult to find the takeaway message at times. I realize this is an in-depth characterization but presenting the data in a clearer fashion that highlights the author's points would be helpful.

Description of the methods need to be improved. Similarly how calculations were made (e.g. pulmonary edema). How were these scores/calculations obtained? Was the person blinded? 

Author Response

The manuscript submitted by Davis et al. is a study into the ability of a variant of SARS-CoV-2 to infect mice. A prior study uncovered SARS-CoV-2 MA10 was able to infect Balbc mice. The current study tested whether C57 mice could be infected and show signs of Covid-19 following inoculation with SARS-CoV-2 MA10. The study is an advancement of the field, albeit incremental. The submitted investigation found two additional mutations present following serial inoculations.

- Thank you for reviewing our manuscript.  We would like to point out that we identified only 1 novel mutation, an Arg to Gly mutation at position 2 of Nsp7, which arose following a single passage in Vero cells.

Strengths

This is a fairly in depth characterization of Covid-19 pathogenesis in mice following inoculation with this virus.

Many aspects of Covid-19 are considered and tested; including age, sex, multiple models, and various organs.

Weaknesses

The limitations should be more detailed to include descriptions of the strengths/weaknesses of the mouse adapted virus. The authors could also describe the advantages/disadvantages for using this virus/mouse to study Covid-19.

- Thank you for pointing out this oversight, we have added text in the introduction to address strengths/weaknesses and advantages/disadvantages of using a mouse adapted SARS-CoV-2.

Illustrating where these mutations are on the virus and the effect on binding to ACE2 would be helpful.

- These mutations were illustrated by us in Figure 2A as well as by Leist and Dinnon in a prior report, which we included as a citation (Reference #3).  Leist and Dinnon also report on the binding of MA10 to Ace2.

It would be helpful to better differentiate R2G from SARS-CoV-2 MA10.

- Figure 1 characterizes the effects of MA10 infection, Figure 2 compares the effects of MA10 vs R2G, and Figures 3 – 8 extensively characterize the effects of R2G infection.  It is unclear what else is needed. We have added text to direct the reader to these data sets for assessment of R2G.

A lot of data is presented, however it was difficult to find the takeaway message at times. I realize this is an in-depth characterization but presenting the data in a clearer fashion that highlights the author's points would be helpful.

- The takeaway messages can be found in the first and last paragraphs of the discussion.  We also note that these comments are in direct contrast with our other 2 reviewers for Vaccines who stated that the data were presented nicely.

Description of the methods need to be improved. Similarly how calculations were made (e.g. pulmonary edema). How were these scores/calculations obtained? Was the person blinded?

- The pathologist was indeed blinded and we apologize for not including this information.  A statement to this effect is in the revised text.  Calculations for the pathology scores were described in the Materials and Methods section and the individual scores are found in the supplementary information: Supplementary Tables 4 – 7.  We have added a statement regarding this to the Materials and Methods section.

Reviewer 3 Report

First, it seems like this manuscript also is under review at Scientific Reports (see https://www.researchsquare.com/article/rs-2194450/v1, assessed on 2022-12-07, 1:30 pm). That issue should be clarified before the submission can be processed further!!

Other than that, the manuscript is well-written and conclusive. That said, I noted some careless mistakes and shortcomings that should be taken care of.  I am listing these few points that should/could be addressed when preparing a revised version of the manuscript.

11.       Mention in the abstract that you’ve used a mouse-adapted virus strain – to prevent false expectations. This is a limitation of the present study and should be discussed as one. Also, the term “accurately” could be deleted in lines 22 and 35.

22.       There’s only one introductory sentence on age- and sex-based differences in SARS-CoV infection in line 289… Yet, I think that it is important to provide some more background on age- and sex-based differences in SARS-CoV infection in your introduction… what are predisposing factors in human COVID? Do other animal models already exist to investigate these? Are these studies in mice dependent on mouse-adapted SARS-CoV-2 strains?

33.       Include comparisons to previous reports on SARS-CoV-2 infection of C57BL/6 mice… like doi: 10.3390/v14050966, doi: 10.1186/s13578-021-00656-8, doi: 10.1101/2021.09.29.462373 and doi: 10.1038/s41392-021-00848-1.

44.       Lines 90f: How many animals were used (indicate n!), were they housed in IVCs?

55.       In the manuscript text, you use superscript numbers as “power” numbers. In the figures, you use x^y. That could be standardized.

66.       Remove “Figure X” from the upper right corner of all figures.

77.       Include missing scale bars in Figs. 1, 4, and in the supplements (where appropriate).

88.       Include stats descriptions in figure legends (e.g. what do asterisks indicate? What do the error bars indicate?) à as in Fig. 8!!

99.       Why did you choose to perform stats on some datasets but not on others (e.g. missing stats in Figs. 1, 2, 3)?

110.   Figs. 4 and S4 are exactly the same… The original Fig. 4 is missing!!

111.   Fig. 7: Panel C doesn’t show RNA-ISH for SCID males, as stated in the legend. Please clarify/revise!

112.   Line 277: Could you provide a more appropriate reference than your ref 20, which has been posted on arXiv in 2020 but doesn’t seem to be peer-reviewed since then?

113.   Line 309: the Wuhan isolate is missing in the “Virus” M&M paragraph.

114.   Line 321: Supplemental Figure 4 does not show dose response for 2 yr-old male and female mice treated with 102, 103, or 104 PFU R2G!

115.   Lines 429f say that Fig. S5 shows tissues of infected mice, while the legend to Fig. S5 says that mice were mock-infected. Please clarify.

116.   Your title highlights that this mouse model phenocopies age- and sex-based differences in human COVID-19… I suggest that you summarize and highlight key aspects of infection and recovery observed in humans vs your mice as a table or summary figure so that the reader can easily compare virus-induced pathology and immune responses in those two species.

117.   The few remaining typos will likely be picked up during copy-editing (e.g. SAR-CoV-2/line 83; version/line 86).

Author Response

Responses to Reviewer #2

First, it seems like this manuscript also is under review at Scientific Reports (see https://www.researchsquare.com/article/rs-2194450/v1, assessed on 2022-12-07, 1:30 pm). That issue should be clarified before the submission can be processed further!!

Thank you for noticing this. We had asked Scientific Reports to retract our manuscript prior to our submission at Vaccines, and they had failed to do so.  It is now retracted.

Other than that, the manuscript is well-written and conclusive. That said, I noted some careless mistakes and shortcomings that should be taken care of.  I am listing these few points that should/could be addressed when preparing a revised version of the manuscript.

We also thank you for your fair critique and careful review of our manuscript.  We comment on specific critiques below.

1. Mention in the abstract that you’ve used a mouse-adapted virus strain – to prevent false expectations. This is a limitation of the present study and should be discussed as one. Also, the term “accurately” could be deleted in lines 22 and 35.

We have added the use of a mouse adapted virus to the abstract.

2. There’s only one introductory sentence on age- and sex-based differences in SARS-CoV infection in line 289… Yet, I think that it is important to provide some more background on age- and sex-based differences in SARS-CoV infection in your introduction… what are predisposing factors in human COVID? Do other animal models already exist to investigate these? Are these studies in mice dependent on mouse-adapted SARS-CoV-2 strains?

We have expanded the comments on age and sex-based differences in the introduction somewhat and will give this topic more focus in the discussion. Also, as mentioned in the introduction, other studies have looked at the role or age or the role of sex, but our study, as far as we can tell, is the first to look at both.

3. Include comparisons to previous reports on SARS-CoV-2 infection of C57BL/6 mice… like #1 (doi: 10.3390/v14050966), #2 (doi: 10.1186/s13578-021-00656-8), #3 (doi: 10.1101/2021.09.29.462373) and #4 (doi: 10.1038/s41392-021-00848-1).

Thank you for pointing out additional interesting studies.  We have added references to studies #2 and #4 above.  We feel that #1 is not relevant as we are not trying to model Beta and Gamma infections but instead reflect infections of the ancestral Wuhan isolate. Likewise, #3 does not yet appear to be peer reviewed.

4. Lines 90f: How many animals were used (indicate n!), were they housed in IVCs?

We apologize for the confusion.  Weights of all animals studied and numbers of mice per cohort in each study can be found in Supplementary Table 1; however, in general, studies of weight loss used 5 mice per cohort and tissue collection studies used 3 mice per cohort unless otherwise noted.  This text was added to the Materials and Methods section of the paper.

5. In the manuscript text, you use superscript numbers as “power” numbers. In the figures, you use x^y. That could be standardized.

We feel the two versions of power annotation are interchangeable and have been used stylistically with intent.

6. Remove “Figure X” from the upper right corner of all figures.

This has been corrected.

7. Include missing scale bars in Figs. 1, 4, and in the supplements (where appropriate).

While scale bars are best, we feel the magnification is sufficient.

8. Include stats descriptions in figure legends (e.g. what do asterisks indicate? What do the error bars indicate?) à as in Fig. 8!!

Thank you for pointing out this oversight.  We have corrected it.

9. Why did you choose to perform stats on some datasets but not on others (e.g. missing stats in Figs. 1, 2, 3)?

Stats have been added where appropriate.

10. Figs. 4 and S4 are exactly the same… The original Fig. 4 is missing!!

Thank you for pointing out this embarrassing oversight.

11. Fig. 7: Panel C doesn’t show RNA-ISH for SCID males, as stated in the legend. Please clarify/revise!

Thank you for pointing this out.  It has been revised and now correctly refers to the comparison of pathology scores between wt and SCiD mice.

12. Line 277: Could you provide a more appropriate reference than your ref 20, which has been posted on arXiv in 2020 but doesn’t seem to be peer-reviewed since then?

The author of reference 20 tells me they are in review, but the data have not yet been published in a peer-reviewed journal.  Unfortunately, it is difficult to get lung biopsies from humans early in the infection process and a better reference has not been found.

13. Line 309: the Wuhan isolate is missing in the “Virus” M&M paragraph.

We are confused by this comment as we did not use the Wuhan isolate, instead we used a mouse adapted virus that was generated from the Wuhan isolate as described in reference #3.

14. Line 321: Supplemental Figure 4 does not show dose response for 2 yr-old male and female mice treated with 102, 103, or 104PFU R2G!

Thank you for finding this typo.  It now reads “Supplemental Figure 3”.

15. Lines 429f say that Fig. S5 shows tissues of infected mice, while the legend to Fig. S5 says that mice were mock-infected. Please clarify.

We apologize for the confusion and have now noted that “older mice in general also had more histopathologic evidence of comorbidities….”

16. Your title highlights that this mouse model phenocopies age- and sex-based differences in human COVID-19… I suggest that you summarize and highlight key aspects of infection and recovery observed in humans vs your mice as a table or summary figure so that the reader can easily compare virus-induced pathology and immune responses in those two species.

We have expanded the discussion section to hopefully provide the requested information.

17. The few remaining typos will likely be picked up during copy-editing (e.g. SAR-CoV-2/line 83; version/line 86).

Thank you for pointing these two out; they have been corrected.

Round 2

Reviewer 3 Report

I appreciate the responses of the authors to my comments and concerns. The revisions they have made have improved the paper and I have no additional comments or concerns.

The revised manuscript is good and I look forward to their future work.

Author Response

Responses to Reviewer

First, it seems like this manuscript also is under review at Scientific Reports (see https://www.researchsquare.com/article/rs-2194450/v1, assessed on 2022-12-07, 1:30 pm). That issue should be clarified before the submission can be processed further!!

Thank you for noticing this. We had asked Scientific Reports to retract our manuscript prior to our submission at Vaccines, and they had failed to do so.  It is now retracted.

Other than that, the manuscript is well-written and conclusive. That said, I noted some careless mistakes and shortcomings that should be taken care of.  I am listing these few points that should/could be addressed when preparing a revised version of the manuscript.

We also thank you for your fair critique and careful review of our manuscript.  We comment on specific critiques below.

  1. Mention in the abstract that you’ve used a mouse-adapted virus strain – to prevent false expectations. This is a limitation of the present study and should be discussed as one. Also, the term “accurately” could be deleted in lines 22 and 35.

We have added the use of a mouse adapted virus to the abstract.

  1. There’s only one introductory sentence on age- and sex-based differences in SARS-CoV infection in line 289… Yet, I think that it is important to provide some more background on age- and sex-based differences in SARS-CoV infection in your introduction… what are predisposing factors in human COVID? Do other animal models already exist to investigate these? Are these studies in mice dependent on mouse-adapted SARS-CoV-2 strains?

We have expanded the comments on age and sex-based differences in the introduction somewhat and will give this topic more focus in the discussion. Also, as mentioned in the introduction, other studies have looked at the role or age or the role of sex, but our study, as far as we can tell, is the first to look at both.

  1. Include comparisons to previous reports on SARS-CoV-2 infection of C57BL/6 mice… like #1 (doi: 10.3390/v14050966), #2 (doi: 10.1186/s13578-021-00656-8), #3 (doi: 10.1101/2021.09.29.462373) and #4 (doi: 10.1038/s41392-021-00848-1).

Thank you for pointing out additional interesting studies.  We have added references to studies #2 and #4 above.  We feel that #1 is not relevant as we are not trying to model Beta and Gamma infections but instead reflect infections of the ancestral Wuhan isolate. Likewise, #3 does not yet appear to be peer reviewed.

  1. Lines 90f: How many animals were used (indicate n!), were they housed in IVCs?

We apologize for the confusion.  Weights of all animals studied and numbers of mice per cohort in each study can be found in Supplementary Table 1; however, in general, studies of weight loss used 5 mice per cohort and tissue collection studies used 3 mice per cohort unless otherwise noted.  This text was added to the Materials and Methods section of the paper.

  1. In the manuscript text, you use superscript numbers as “power” numbers. In the figures, you use x^y. That could be standardized.

We feel the two versions of power annotation are interchangeable and have been used stylistically with intent.

  1. Remove “Figure X” from the upper right corner of all figures.

This has been corrected.

  1. Include missing scale bars in Figs. 1, 4, and in the supplements (where appropriate).

While scale bars are best, we feel the magnification is sufficient.

  1. Include stats descriptions in figure legends (e.g. what do asterisks indicate? What do the error bars indicate?) à as in Fig. 8!!

Thank you for pointing out this oversight.  We have corrected it.

  1. Why did you choose to perform stats on some datasets but not on others (e.g. missing stats in Figs. 1, 2, 3)?

Stats have been added where appropriate.

  1. Figs. 4 and S4 are exactly the same… The original Fig. 4 is missing!!

Thank you for pointing out this embarrassing oversight.

  1. Fig. 7: Panel C doesn’t show RNA-ISH for SCID males, as stated in the legend. Please clarify/revise!

Thank you for pointing this out.  It has been revised and now correctly refers to the comparison of pathology scores between wt and SCiD mice.

  1. Line 277: Could you provide a more appropriate reference than your ref 20, which has been posted on arXiv in 2020 but doesn’t seem to be peer-reviewed since then?

The author of reference 20 tells me they are in review, but the data have not yet been published in a peer-reviewed journal.  Unfortunately, it is difficult to get lung biopsies from humans early in the infection process and a better reference has not been found.

  1. Line 309: the Wuhan isolate is missing in the “Virus” M&M paragraph.

We are confused by this comment as we did not use the Wuhan isolate, instead we used a mouse adapted virus that was generated from the Wuhan isolate as described in reference #3.

  1. Line 321: Supplemental Figure 4 does not show dose response for 2 yr-old male and female mice treated with 102, 103, or 104PFU R2G!

Thank you for finding this typo.  It now reads “Supplemental Figure 3”.

  1. Lines 429f say that Fig. S5 shows tissues of infected mice, while the legend to Fig. S5 says that mice were mock-infected. Please clarify.

We apologize for the confusion and have now noted that “older mice in general also had more histopathologic evidence of comorbidities….”

  1. Your title highlights that this mouse model phenocopies age- and sex-based differences in human COVID-19… I suggest that you summarize and highlight key aspects of infection and recovery observed in humans vs your mice as a table or summary figure so that the reader can easily compare virus-induced pathology and immune responses in those two species.

We have expanded the discussion section to hopefully provide the requested information.

  1. The few remaining typos will likely be picked up during copy-editing (e.g. SAR-CoV-2/line 83; version/line 86).

Thank you for pointing these two out; they have been corrected.
